# Differences in Aortopathy in Patients with a Bicuspid Aortic Valve with or without Aortic Coarctation

**DOI:** 10.3390/jcm9020290

**Published:** 2020-01-21

**Authors:** Anthonie Duijnhouwer, Allard van den Hoven, Remy Merkx, Michiel Schokking, Roland van Kimmenade, Marlies Kempers, Arie van Dijk, Menko-Jan de Boer, Jolien Roos-Hesselink

**Affiliations:** 1Department of Cardiology, Radboud University Medical Center, 6500HB Nijmegen, The Netherlands; Remy.Merkx@radboudumc.nl (R.M.); Roland.vanKimmenade@radboudumc.nl (R.v.K.); Arie.vanDijk@radboudumc.nl (A.v.D.); menkojan@gmail.com (M.-J.d.B.); 2Department of Cardiology, Erasmus Medical Center, 3000CA Rotterdam, The Netherlands; a.vandenhoven@erasmusmc.nl (A.v.d.H.); j.roos@erasmusmc.nl (J.R.-H.); 3Department of Pediatric Cardiology, Amalia Children’s Hospital, Radboud University Medical Center, 6500HB Nijmegen, The Netherlands; Michiel.Schokking@radboudumc.nl; 4Department of Genetics, Radboud University Medical Center, 6500HB Nijmegen, The Netherlands; Marlies.Kempers@radboudumc.nl

**Keywords:** Bicuspid aortic valve, aortic dilation, aortic dissection, aortic coarctation, congenital heart disease, cardiac magnetic resonance imaging

## Abstract

Objective: The combination of aortic coarctation (CoA) and bicuspid aortic valve (BAV) is assumed to be associated with a higher risk of ascending aortic dilatation and type A dissection, and current European Society of Cardiology (ESC) guidelines advise therefore to operate at a lower threshold in the presence of CoA. The aim of our study is to evaluate whether the coexistence of CoA in BAV patients is indeed associated with a higher risk of ascending aortic events (AAE). Methods: In a retrospective study, all adult BAV patients visiting the outpatient clinic of our tertiary care center between February 2003 and February 2019 were included. The primary end point was an ascending aortic event (AAE) defined as ascending aortic dissection/rupture or preventive surgery. The secondary end points were aortic dilatation and aortic growth. Results: In total, 499 BAV patients (43.7% female, age 40.3 ± 15.7 years) were included, of which 121 (24%) had a history of CoA (cBAV). An aortic event occurred in 38 (7.6%) patients at a mean age of 49.0 ± 13.6 years. In the isolated BAV group (iBAV), significantly more AAE occurred, but this was mainly driven by aortic valve dysfunction as indication for aortic surgery. There was no significant difference in the occurrence of dissection or severely dilated ascending aorta (>50 mm) between the iBAV and cBAV patients (*p* = 0.56). The aortic diameter was significantly smaller in the cBAV group (30.3 ± 6.9 mm versus 35.7 ± 7.6 mm; *p* < 0.001). The median aortic diameter increase was 0.23 (interquartile range (IQR): 0.0–0.67) mm/year and was not significantly different between both groups (*p* = 0.74). Conclusion: Coexistence of CoA in BAV patients was not associated with a higher risk of aortic dissection, preventive aortic surgery, aortic dilatation, or more rapid aorta growth. This study suggests that CoA is not a risk factor in BAV patients, and the advice to operate at lower diameter should be reevaluated.

## 1. Introduction

Bicuspid aortic valve (BAV) is the most common congenital heart defect with a prevalence of 0.5–1.3% [1]. In about half of all BAV patients, ascending aortic dilation develops, predisposing for life-threatening complications such as aortic dissection [2]. BAV can occur isolated (iBAV), but also in combination with additional heart defects or in the context of a syndrome.

Depending on age groups studied, the prevalence of aortic coarctation (CoA) in BAV patients varies between 22% and 36% and is increased in younger age groups [2,3,4,5]. Whether there is a difference in prevalence of aortopathy between BAV patients with (cBAV) or without a CoA remains unknown.

After CoA correction, many patients suffer from hypertension, which is a known risk factor for aortic dissection and higher mortality, especially in BAV patients [4,6]. In a pediatric cohort of BAV patients, CoA was associated with smaller ascending aortic diameters, while in adults, contradicting studies about CoA in BAV patients have been published, with some showing a higher risk of ascending aortic events (AAE) when CoA was present in these patients, while others found no relation [4,7,8].

The aortopathy in BAV patients is generally assumed and feared, and although the incidence of type A aortic dissections is low, it is clearly higher compared with the general population with BAV patients having an estimated 6 times higher risk [9]. Although only a few studies have investigated the impact of CoA in BAV patients on the incidence of aortopathy, current ESC guidelines on aorta pathology advise to consider preventive aortic surgery at a lower aortic diameter in BAV patients when a history of CoA is present [10].

The aim of this study is to investigate whether the coexistence of CoA in BAV patients is associated with ascending aortic events (AAE), aortic dilatation and aortic diameter increase.

## 2. Method

Medical ethical committee region Arnhem–Nijmegen approved this study under file number CMO: 2017-3599.

In this retrospective study, we included all BAV patients who visited the adult outpatient clinic between February 2003 and February 2019. Exclusion criteria were presence of an associated complex congenital heart disease (e.g., tetralogy of Fallot) or a hereditary thoracic aorta disease (e.g., Marfan syndrome) (Flow diagram Figure 1) and incomplete data. Hemodynamic unimportant small defects or corrected defects were not excluded (e.g., ventricular septal defect, atrial septal defect, persisting left cava vein, persisting ductus arteriosus, subaortic membrane, abnormal pulmonary venous return). BAV presence was diagnosed on echocardiography or on cardiac magnetic resonance imaging (CMR) or by the surgeon during aortic valve surgery. Patients with a hemodynamic important CoA (gradient >20 mmHg across the isthmus stenosis invasively measured or hypertension in presence of a >50% aortic narrowing compared with aortic diameter at the diagram) were assigned to the BAV with CoA group (cBAV). All patients with a hemodynamic important CoA underwent an intervention (surgery or percutaneous). Patients without CoA were assigned to the isolated BAV group (iBAV).

Following our standard protocol, during each visit all patients underwent an electric cardiography (ECG) and echocardiography. Cardiologists with expertise in congenital heart disease and echocardiography evaluated the echocardiographic images. Quantification of valve dysfunction severity was done according to current guidelines [11,12]. Moderate to severe stenosis or regurgitation was defined as significant aortic valve dysfunction.

BAV morphology was classified following the Sievers classification [13]. Frequency of follow-up was determined depending on the severity of valve dysfunction and other relevant comorbidities. In most cases, this was either annually or biannually. Hypertension was defined as a blood pressure above 140/90 mmHg on several measurements, all these patients were treated with antihypertensive medication [14].

The primary end point of this study was an ascending aortic event (AAE), defined as the occurrence of acute dissection of the ascending aorta or the occurrence of (preventive) ascending aortic surgery. The secondary end points were aortic diameter on CMR or CT at age >16 years of age and aortic growth during follow-up using CMR or CT during adulthood.

### 2.1. Advanced Aortic Imaging

Aortic diameters were measured on advanced imaging by experienced radiologists. Cardiac Magnetic Resonance Imaging (CMR) was preferably used and in case CMR was not possible or contra-indicated, an ECG-triggered CT scan was performed. The ascending aorta was measured in the axial plane, during diastolic phase. In both modalities, the inner edge to inner edge method was used to measure the ascending aortic diameter at the height of the right pulmonary branch [15]. Ascending aortic diameter was defined as dilated when the ascending aorta >40 mm or aortic size index >20 mm/m^2^ [10]. Frequency of advanced imaging was based on indication, but performed at least once during the first visit to the adult outpatient clinic and typically repeated every 5 years.

The aortic diameter change (in mm) between two CMR/CT ascending aortic measurements was divided by the time between the two measurements, which had to be at least one year apart. For every successive scan, the aortic diameter change was calculated. This implies that multiple “means” could be present in one patient, in that case the means were added up and divided by the number of means for that specific patient, resulting in one mean aortic diameter change per patient.

### 2.2. Statistical Analysis

Statistical analysis was performed using Statistical package for social sciences, version 25 for Windows (SPSS, Chicago, IL, USA). Results are expressed as mean ± standard deviation or as median 25% and 75% interquartile range (IQR) if the distribution was skewed. A *p*-value <0.05 was defined to be statistically significant. The independent samples T-test was used to compare means between groups. In case of a skewed distribution, the Mann–Whitney U test was used. To evaluate a significant difference between proportions, a chi-square test was used. Univariate and multivariate logistic regression was used to correct for important determinants of AAE, for example, age, aortic valve dysfunction, and hypertension.

## 3. Results

### 3.1. Baseline

A total of 499 BAV patients were identified, of which 121 (24.2%) were diagnosed to have CoA (cBAV). Baseline characteristics of all included BAV patients are shown in Table 1.

The median age of coarctation repair was 10 month (IQR: 2 months; 7.9 years), of which 107 had a surgical repair. Balloon angioplasty of descending aorta was performed in 14 patients, at a median age of 13.5 (IQR: 3.8; 31.5) years, and a stent was implanted in 12 of them.

The prevalence of aortic valve regurgitation was significant higher in the iBAV group (*p* = 0.006). There was no age difference between patients with (41.6 ± 16.2 years) or without (40.4 ± 15.7 years) a significant aortic valve regurgitation in total group or in the iBAV group (42.6 ± 16.1 years in no significant versus 44.0 ± 16.9 years in significant aortic valve regurgitation).

### 3.2. Events Analysis

Table 2 shows the ascending aortic events (AAE) for the total group and iBAV and cBAV groups separately. The mean age at AAE was not significantly different between both groups. Aortic dissection occurred in two iBAV patients and in one cBAV patient, who was also a Turner patient; none of these three patients survived. No aortic ruptures occurred. There was a significant difference in AAE between iBAV group and cBAV group (*p* = 0.016). This was mainly driven by aortic valve dysfunction as primary indication for aortic valve replacement in combination with ascending aorta replacement. When patients with aortic valve dysfunction as primary indication were excluded, there was no significant difference in prevalence of AAE between iBAV and cBAV groups (*p* = 0.743). There was no significant difference (*p* = 0.711) in the prevalence of high-risk BAV patients (ascending aortic diameter >55 mm and ascending aortic dissection).

Figure 2 illustrates the age distribution at which the AAE occurred for both groups. For each age group, the patients with an AAE were divided by the total patients in this age group. In the age groups <20 years, 30–40 years, and 50–60 years, no AAE were observed. In the other age groups, the percentages were comparable. Table 3 show the univariate and multivariate analysis of AAE. Due to the low event rate, a limited number of variables could be tested (CoA, hypertension, age, aortic regurgitation).

### 3.3. Ascending Aortic Diameter on CMR/CT

At least one CMR/CT was available in adulthood in 416 BAV (83%) patients. The mean aortic diameter at first CMR/CT was 34.4 ± 7.8 mm, at a median age of 27.2 (20.2–43.1) years. There was a significant difference in age between the 312 patients in the iBAV group (median age of 30.5 (21.0–47.5)) and the 104 patients in the cBAV group (median age 21.9 (18.4–30.8)) (*p* < 0.001) and a significant difference in aortic diameter between iBAV (mean diameter of 35.7 ± 7.6 mm) and cBAV group (mean diameter 30.3 ± 6.9 mm) (*p* < 0.001). The aortic diameter in the cBAV group was still significantly smaller (coefficient = −3.42; *p* < 0.001) after correcting for age (coefficient = 0.23; *p* < 0.001). The mean aortic size index (absolute diameter divided by body surface area) was 18.6 ± 4.0 mm/m^2^ and was significant lower in the cBAV group compared with the iBAV group (16.2 ± 3.7 versus 19.4 ± 3.9 mm/m^2^; *p* < 0.001). After correcting for age, cBAV patients had still a smaller aortic size index (coefficient of −2.6; *p* < 0.001).

Ascending aortic dilatation defined as a diameter >40 mm was significant more prevalent in the iBAV group compared with the cBAV group (31.1% versus 9.8%; *p* < 0.001).

Ascending aorta dilatation when defined as aortic size index >20 mm/m^2^ was also significant more prevalent in the iBAV group (45.0% versus 13.7%; *p* < 0.001). In Turner women, the aortic size index was significantly larger compared with the rest of the group (19.9 mm/m^2^ versus 18.6 mm/m^2^; *p* = 0.043). cBAV was present in 8.6% of all Turner women and in 1.2% of all BAV patients.

All patients with an ascending aortic diameter >55 mm were operated. There was no significant difference between the iBAV and cBAV groups (2.6% versus 1.0%; *p* = 0.46) in the prevalence of nonoperated patients with an ascending aortic diameter >50 mm.

### 3.4. Aortic Diameter Increase during Follow-up

In 162 patients (49 cBAV, 30%), more than one CMR/CT was available. The mean follow-up time was 6.0 ± 2.9 years, the median ascending aortic diameter change was 0.23 (0.0–0.67) mm/year. Figure 3 shows the boxplot of the median ascending aortic diameter change of the iBAV group (median age at first CMR/CT 28.6 (20.3–45.0) years) and the cBAV group (median age at first CMR/CT 21.2 (18.0–29.9) years.

There was no significant difference (*p* = 0.74) in median aortic diameter change between the iBAV group (0.24 (−0.02–0.76) mm/year; *n* = 114) and the cBAV group (0.20 (0.00–0.57) mm/year; *n* = 49). None of the patients had an aortic diameter increase of >3 mm/year.

In the 162 patients with two or more CMR/CT scans (follow-up group), 22.7% had hypertension, this was not significantly different in the iBAV and cBAV groups (20.2% versus 28.6%; *p* = 0.241). Of the 14 patients with an ascending aortic diameter of >50 mm at first CMR/CT (1 cBAV patient), four patients were operated before the second CMR/CT was performed. The remaining 10 patients had a median ascending aortic change of −0.13 mm (−0.85–0.14).

## 4. Discussion

The presented data in this large study of relative young BAV patients suggest that patients with and without aortic coarctation are both similarly associated with aortic events and show a comparable increase in ascending aortic diameter. The current idea is that large ascending aortic diameters are associated with a higher risk for future AAE. In this study, ascending aortic diameter was significantly higher in the iBAV group, probably due to higher age. Nevertheless, the prevalence of preventive ascending aortic surgery was comparable to the cBAV group, when the primary indication was ascending aortic dilation.

The current ESC aortic guideline and ESC valve guideline both state that ascending aortic surgery should be advised (class IIa, level C) at a lower ascending aortic diameter (50 mm instead of 55 mm) in BAV patients with a (history of) CoA [10,16]. In this study, no evidence to substantiate this recommendation is found. In fact, the results points towards a more favorable course in cBAV patients, and therefore the recommendation of the ESC should be revisited.

Eleid et al. presented a small cohort of aortic dissection in BAV patients, in which they concluded that cBAV was associated with dissection, since they found that 23% of the BAV patients had a CoA [5]. This conclusion is debatable because the normal prevalence of CoA in BAV is about 23% and, therefore, represents a normal distribution of iBAV and cBAV [5].

Oliver et al. described a group of BAV patients (*n* = 341), in which they found that the coexistence of CoA was associated with more ascending aortic events. They defined an event as a dissection or rupture of the sinus of Valsalva or an aortic dilatation > 55 mm [4] They experienced the same problem as the current study of a low event rate and a younger-aged cBAV group with a mean age of 18 (16–23) years [4]. The difference with the current study can partly be explained by a slightly different AAE definition (in this study, also preventive aortic surgery patients were included), which could have caused a higher event rate in the current study.

Michelena et al. reported 416 BAV patients, in whom they found no increased AAE in cBAV patients during a mean follow-up time of 16 ± 7 years [6]. Tzemos et al. showed that a history of CoA was associated with a lower event rate [17].

The fact that this study and other studies did not find significant differences in AAE between iBAV and cBAV suggest that the recommendation of earlier preventive surgery in cBAV patients needs to be reconsidered [6,17].

Aortic diameter is the most important parameter on which the indication for preventive aortic surgery is based, although some other factors are important too, such as hypertension [18]. In the current study, as in other studies, it was observed that coexistence of CoA was associated with smaller aortic diameters [6,19,20]. This implies that cBAV patients are less likely to develop an aortic aneurysm needing preventive aortic surgery. On the other hand, a higher prevalence of hypertension was observed in the cBAV group, and hypertension has been associated with AAE, especially dissection [21]. The development of hypertension could be induced by the relative aortic hypoplasia after CoA correction. The definition of aortic hypoplasia is guided by the application of the Z-score for the aortic arch, but at which z-score hypertension develops is not clear.

Rapid aortic growth (>3–5 mm/year) is a risk factor for future AAE and is an indication to consider preventive surgery [10].

The ascending aortic diameter increase was slow (0.20–0.23 mm/year) and not different between the iBAV and cBAV groups in the current study, and this is in line with previous reports (Oliver 2009). This suggests that CoA is not associated with increased growth rates in BAV patients and thus probably not associated with a higher risk of AAE, also after longer follow-up.

In this study, only patients with a BAV were analyzed, and whether these data also apply to patients with an aortic coarctation and a normal aortic valve was not investigated.

### Limitations

Several potential limitations must be noted. All limitations associated with retrospective research apply to this study. Due to the low number of events, no hard conclusions can be drawn from this paper, only suggestions, but for now it is the best we have. Our study population consisted of patients receiving care in a specialized center and may therefore be less generalizable. The age difference between the two groups was partly caused by referral bias, since BAV was, especially in the older patients, more often diagnosed during aortic surgery. This problem was largely corrected by conducting a multivariate analyses in which we corrected for age. A potential limitation is also caused by the fact that CMR/CT was not performed in every patient at a fixed time interval. Finally, we did not evaluate specific aortic dilatation patterns.

## 5. Conclusions

This study suggests that the ascending aortic event rate is not different between BAV patients with or without a history of aortic coarctation. Implying that a difference in indication for ascending aortic surgery is not justified in BAV patients based on a history of aortic coarctation alone.

## Figures and Tables

**Figure 1 jcm-09-00290-f001:**
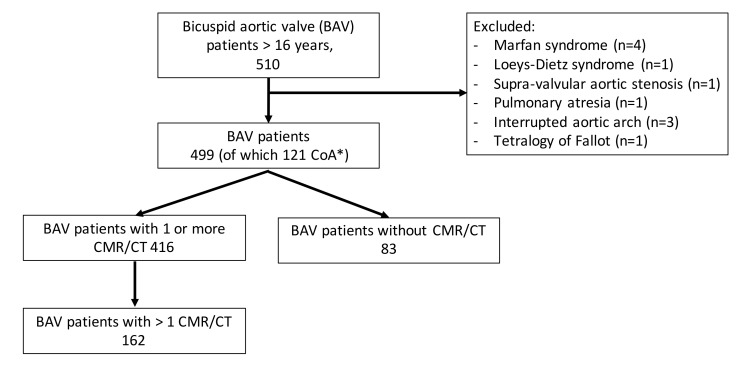
Inclusion flow chart. Flow chart describing the reason for inclusion and exclusion of all eligible BAV patients in current study; CMR/CT, cardiac magnetic resonance imaging and computed tomography; * aortic coarctation (CoA).

**Figure 2 jcm-09-00290-f002:**
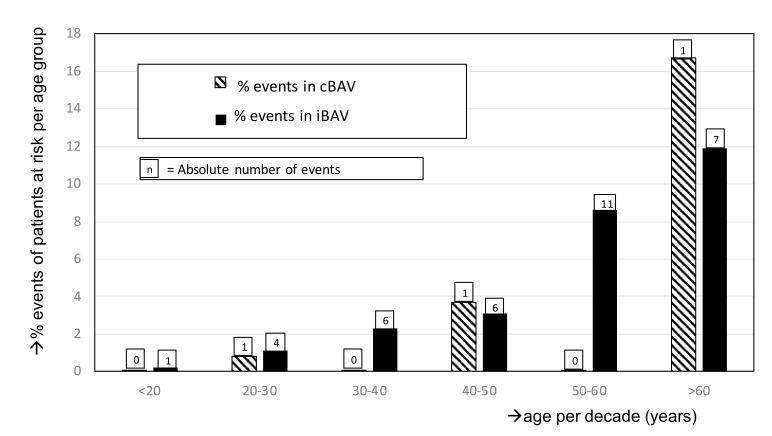
Aortic events in 499 BAV patients, as percentage of total in each age group for iBAV group and cBAV group. The vertical axis represents the percentage of events, calculated as the total patients at risk per specific age group of iBAV and cBAV groups.

**Figure 3 jcm-09-00290-f003:**
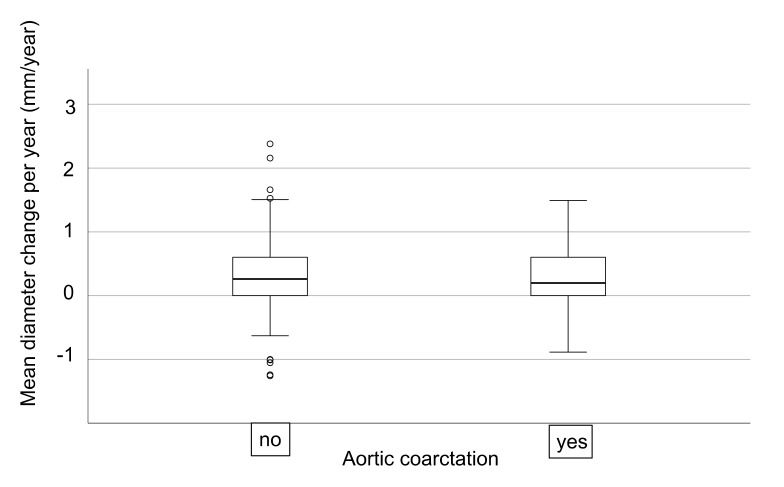
Boxplot of mean ascending aortic diameter change (mm/year) for iBAV and cBAV groups, calculated from 162 unique patients with more than one CMR/CT.

**Table 1 jcm-09-00290-t001:** Baseline characteristics of BAV patients with and without coarctation.

	All BAV Patients (*n* = 499)	iBAV (*n* = 378)	cBAV (*n* = 121)	*p*-Value
Female (%)	43.7	44.2	42.1	0.695
Age at inclusion ^§^ (years)	34.1 ± 15.3	36 ± 15.9	28.2 ± 11.6	<0.001
Age at end of study (years)	40.3 ± 15.7	42.5 ± 16.1	33.6 ± 12.2	<0.001
Weight (kg)	72.0 (62.0–82.0)	72.0 (62.0–82.0)	71 (60.0–81.3)	0.519
Height (cm)	172.8 ± 12.3	172.7 ± 12.6	173.5 ± 11.4	0.482
Turner (%)	14.5	16.9	5.0	0.001
Hypertension (%)	19.8	17.2	28.9	0.005
Smoking (%)	10.7	10.4	11.6	0.713
Diabetes (%)	2.0	1.9	2.5	0.683
Hypercholesterolemia (%)	7.6	9.4	3.3	0.031
Other congenital defect * (%)	22.9	17.2	38.8	<0.001
Aortic sinus of Valsalva (mm) **	33.4 ± 8.9	33.8 ± 6.2	32.3 ± 5.8	0.026
Ascending aorta (mm) **	34.4 ± 7.2	35.3 ± 7.2	31.6 ± 6.5	<0.001
Aortic regurgitation moderate/severe (%) **	17.9	20.7	9.2	0.006
Aortic valve stenosis moderate/severe (%) **	5.7	5.7	5.5	0.924

^§^ age at first visit of the adult outpatient clinic; * persisting left cava vein (*n* = 4), persisting ductus arteriosus (*n* = 16), ventricular septal defect (*n* = 23), atrial septal defect (*n* = 10), subaortic membrane (*n* = 6), partial abnormal pulmonary venous return (*n* = 4); ** echocardiographic measurement at first visit adult outpatient clinic; BAV, bicuspid aortic valve; iBAV, isolated bicuspid aortic valve; cBAV, aortic coarctation bicuspid aortic valve.

**Table 2 jcm-09-00290-t002:** Characteristics of BAV patients with an aortic event.

	All	iBAV	cBAV	*p*-Value
Number of events (% of event in group) *	38 (7.6%)	35 (9.3%)	3 (2.5%)	0.016
Age at event	48.0 ± 14.4	48.2 ± 14.3	46.4 ± 19.2	0.821
Type of event				
Dissection Rupture (% of total in group) *	3 (0.06%) 0	2 (0.5%) 0	1 (0.8%) 0	NA
Surgery Indication	35	33 (8.7%)	2 (1.7%)	0.007
Valvular dysfunction	21 (4.2%)	21 (5.6%)	0	0.004
(% of total in group) *				
AA > 45 mm	13 (2.6%)	13 (3.4%)	0	
AA > 50 mm	8 (1.6%)	8 (2.1%)	0	
Aortic dilatation	13 (2.6%)	11 (2.9%)	2 (1.7%)	0.743
(% of total in group) *				
AA > 50 mm	6 (1.2%)	6 (1.6%)	0	
AA > 55 mm	7 (1.4%)	5 (1.3%)	2 (1.7%)	
-unknown	1	1	0	

NA: not applicable; AA = ascending aorta diameter; * events divided by total patients in the group.

**Table 3 jcm-09-00290-t003:** Predictors of acute aortic events.

	Univariate Analysis Odds Ratio (95% CI)	*p*-Value	Multivariate Analysis, Odds Ratio (95% CI)	*p*-Value
Age	1.063 (1.041–1.086)	<0.001	1.054 (1.029–1.080)	<0.001
Aortic coarctation	0,249 (0.075–0.825)	0.023	0.410 (0.117–1.441)	0.164
Hypertension	2.664 (1.316–5.391)	0.006	1.329 (0.589–2.998)	0.494
Moderate to severe aortic regurgitation	1.364 (0.922–2.017)	0.120

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
