# Peer review of "Differences in Aortopathy in Patients with a Bicuspid Aortic Valve with or without Aortic Coarctation"

_jcm, 2020, doi:10.3390/jcm9020290_

Round 1

Reviewer 1 Report

This study deals with an overall interesting topic for cardiologists and aimed at clarifying whether aortic coarctation would influence the risk of adverse aortic events in adults with bicuspid aortic valve.

The study was retrospective, 499 patients with BAV were included, of whom 121 also had had a coarctation.

Patients without coarctation were older than those with and showed higher aortic diameters. Apart from more frequent indication for aortic valve surgery due to valve dysfunction in the isolated BAV group, there was no difference in terms of aortic events such as dissection in comparison to the BAV group with coarctation.

General comments:

Authors compared patients with isolated aortic valve with a group associating bicuspid aortic valve and coarctation of the aorta.
It is probably worth to compare both groups but the fact that both patient groups belong probably to 2 different clinical and pathophysiological entities must be highlighted. 
As an example, patients with coarctation had more often additional structural defects that may have influenced aortic growth (VSD, ASD).
The role of a putative aortopathy in both groups that may have a different pysiopathology must be considered. The retrospective design is in my opinion acceptable for this kind of observationnal study.
However, inclusion and exclusion critera must be stated clearly.
An important bias in the study design may be the fact that a number of patient files were not available and that not all events were correctly documented as authors state in the limitation section.

Specific Comments
1. Title: in the current form, the title implies that a group of patients with the same anomaly (BAV) is analyzed with respect to the presence or not of an additionnal anomaly (coarctation). This does not take into consideration the presence of 2 different entities. Introduction: the role of the arteriopathy in both BAV and coarctation must be explained in this section.

2. Patients and methods: 
a) Please provide clear inclusion and exclusion criteria. A further exclusion criteria should also be the absence of complete patient file.
b) More details about the associated lesions (ASD, VSD, PAPVR...) should be provided (unimportant defects is not precise enough) as should be precisely given the technique used for coarctation repair (surgical/interventional/with/without foreign material/what kind of material) and the age at intervention(s).
In patients with coarctation, some had probably associated relative aortic hypoplasia. This must be clarified.
c) Authors excluded patients with Marfan or Loeys-Dietz syndrom. How were these conditions definitively excluded in the patients entered in the study? Had all genetic testing?
d) Treatment by b-blockers or vasodilators (sartan) must be specified and probably entered in the multi-variate analysis.
e) statistical analysis: how was distribution tested? Why is the p-value for consideration of significance set at < or = 0,05 and not < 0,05?
How were the data corrected for age (see result section)?
Since patients with BAV without coarctation were older than the others, did authors consider to age-match them?

3. Results
a) Please specify how the results are presented when the mean value is given
b) Tables and figures, inlcuding legends are not clearly readable and merit to be re-formated. Several mistakes in figure 1.
c) I did not found the results of the multivariable analysis
d) How were data corrected for age (sentence 144)?
e) authors state (sentence 118)  that in coarctation group, there was signifivanly more patients with aortic diameter > 55 mm, but the p-value shown is not signifivant.

4. Discussion
Authors should highlight was their paper brings as new information and be cautious about their conclusions, given the important limitations of their study design.
Discussing the pathophysiology of aortic dilation and dissection is of interest.

Reviewer 2 Report

Authors conducted a retrospective study on patients diagnosed with bicuspid aortic valve which aim was to compare a risk for ascending aortic events (AAE) between isolated valve anomaly (iBAV) and the one coexisting with aortic coarctation (cBAV). The study has a practical value because of usually fatal outcomes of ascending aortic dissection or ruptures. Currently, preventive surgical intervention is guided by ascending aorta diameter. ESC recommends 5 mm lower aortic diameter threshold for surgery in cBAV vs. iBAV. The size of the study groups was substantial and over 80% subjects had CMR or CT imaging with complementary echocardiography, which validate the methodology of the study.

Major comments:

The endpoint of the study impacting conclusions the most was decision on surgical intervention. This was much more common in iBAV group. When compared with cBAV, valvular disfunctions but not aortic dilatation were driving this decisions. Surgery was done much more common in the iBAV group aged 30-60 years, 39% of whom did not have aortic dilatation (< 50 mm). This finding should be better explained and discussed. Quantification of valve disfunction mentioned on page 3 line 73 should be included in the results. In my opinion, endpoint based on elective surgery can obfuscate the study outcome because of many confounders like overall feasibility of surgery, presence of symptomatic aortic valve stenosis or regurgitation, progress of leaflet calcification etc. In the baseline characteristics table is striking low incidence of coarctation of aorta in Turner syndrome patients. Were these Turner syndrome cases scrupulously checked for surgical intervention during their infancy. Perhaps PDA was not only ligated but also aortoplasty was done. Conclusions on the increase of the diameter of aorta in follow-up should be validated by accuracy of the measurements. Author could provide some estimations of errors by the methods used.

Reviewer 3 Report

Duijnhouwer et al need to be congratulated for studying an important and less well studied association of bicuspid aortic valve and coarctation of aorta with aortic events. their methods are sound but results and conclusions need clarification. 

The biggest take home message from this study could be that isolated bicuspid aortic valve and bicuspid valve with coarctation are both similarly associated with increase in aortic sizes/ events over time and ESC guidelines give a recommendation with little evidence and need to be revisited. Authors also report that iBAV is more commonly associated with a large aortic diameter? this finding is contradicted in line 119. could the authors clarify did the authors do a multivariate adjusting for Turner's syndrome. or only age was adjusted for in their analyses. with different frequency of Turner's this needs clarification.  I think if any single imaging modality is available in all patients, authors could look at measurements over time and perhaps share meaningful insight regarding this cohort. from what authors write, only 10 patients were used for figure 3 calculations. could the authors show serial aortic root diameters from TTEs and see them over time to support similar findings. This would increase their number and perhaps plausibility of the growth rate they report.  study findings need to be very clearly written as it is very hard to decipher what the authors found.  manuscript needs major formatting, needs to be written clearly and in a more reader friendly manner.  overall clinical event rates are low to draw meaningful clinical conclusions.  authors make no mention of degree of valvular dysfunction (stenosis or regurgitation) or any adjustments there of in their analysis. One can argue iBAV may have been older and hence had a greater degree of valvular stenoses etc to affect findings. 

Round 2

Reviewer 1 Report

The authors adressed most of the comments adequately and improved significantly their manuscript.

There are still spelling/grammatical mistakes in figure 1 (Tetralogy instead of Tetrology ...)

Figure legend for figure 1 is needed.
S146: please precise that APVR was partial.

Figure 3: the legend of absciss and ordinate is hardly readable.

Discussion, 1rst paragraph: please review the last sentence.

S236: definition of aortic arch hypoplasia is guided by the application of the Z-score for that structure

S242: the citation should be numbered

Reviewer 2 Report

Authors responded to my comments, except the last one on the accuracy of measurements of the aortic diameter. If this was not assessed in the current study, perhaps a reference publication might help.

I find the revised version much improved. Of minor issues, on p. 4, line 126 the sentence on ballon plasty requires rewriting because of the subjects redundance.

Reviewer 3 Report

thank you for answering and addressing all the concerns. 
